Prognostic factors and treatment of neuroendocrine tumors of the uterine cervix based on the FIGO 2018 staging system: a single-institution study of 172 patients

http://orcid.org/0000-0003-4234-2028 Chen Jian 1
Sun Yang 1
Chen Li 1
Zang Lele 1
Lin Cuibo 1
Lu Yongwei 1
Lin Liang 1
Lin An 1
Dan Hu 2
Chen Yiyu 2
He Haixin 1 63804657@qq.com
1 Department of Gynecological-Surgical Oncology, Fujian Medical University Cancer Hospital, Fujian Cancer Hospital , Fuzhou, Fujian , China
2 Department of Pathology, Fujian Medical University Cancer Hospital, Fujian Cancer Hospital , Fuzhou, Fujian , China
Bartolini Barbara
Electronic publication date: 2021 Jul 6
Publication date: 2021
Volume: 9
Electronic Location ID: e11563
Received 2020 Nov 13; Accepted 2021 May 14
Copyright: © 2021 Chen et al.
Copyright year: 2021
Copyright holder: Chen et al.
License: This is an open access article distributed under the terms of the Creative Commons Attribution License, which permits unrestricted use, distribution, reproduction and adaptation in any medium and for any purpose provided that it is properly attributed. For attribution, the original author(s), title, publication source (PeerJ) and either DOI or URL of the article must be cited.
License URL: https://creativecommons.org/licenses/by/4.0/

Keywords: Neuroendocrine carcinoma, Prognostic factors, Uterine cervix, Radical surgery, Adjuvant therapy, Chemotherapy, Radiotherapy

Funding: Fujian Provincial Health Technology Project 2020CXA014 This work was supported by the Fujian Provincial Health Technology Project (Grant Number: 2020CXA014). The funders had no role in study design, data collection and analysis, decision to publish, or preparation of the manuscript.

==============================
Objective

This study aimed to explore the prognostic factors and outcomes of patients with neuroendocrine tumors (NETs) of the cervix and to determine appropriate treatment.

Methods

A single-institution retrospective analysis of 172 patients with NETs was performed based on the new International Federation of Gynecology and Obstetrics (FIGO 2018) staging system.

Results

Among the 172 eligible patients, 161 were diagnosed with small cell neuroendocrine carcinoma (SCNEC), six with large cell neuroendocrine carcinoma, four with typical carcinoid tumors and one with SCNEC combined with an atypical carcinoid tumor. According to the FIGO 2018 staging guidelines, 31 were stage I, 66 were stage II, 57 were stage III, and 18 were stage IV. The 5-year survival rates of patients with stage I–IV disease were 74.8%, 56.2%, 41.4% and 0%, respectively. The 5-year progression-free survival rates of patients with stage I–IV disease were 63.8%, 54.5%, 30.8% and 0%, respectively. In the multivariate analysis, advanced FIGO stage, large tumor and older age were identified as independent variables for 5-year survival in patients with stage I–IV disease. FIGO stage, tumor size and para-aortic lymph node metastasis were independent prognostic factors for 5-year progression-free survival in patients with stage I–IV disease. For the patients receiving surgery (n = 108), tumor size and pelvic lymph node metastasis were independent prognostic factors for 5-year survival, and pelvic lymph node metastasis for 5-year progression-free survival. In stage IVB, at least six cycles of chemotherapy (n = 7) was associated with significantly better 2-year OS (83.3% vs. 9.1%, p < 0.001) and 2-year PFS (57.1% vs. 0%, p = 0.01) than fewer than six cycles of chemotherapy(n = 11).

Conclusion

Advanced FIGO stage, large tumor, older age and lymph node metastasis are independent prognostic factors for NETs of the cervix. The TP/TC and EP regimens were the most commonly used regimens, with similar efficacies and toxicities. Standardized and complete multimodality treatment may improve the survival of patients with NETs.

Introduction

Neuroendocrine cervical tumors (NETs) are a rare but highly aggressive form of cervical cancer (Gardner, Reidy-Lagunes & Gehrig, 2011; Satoh et al., 2014). NETs are divided into four categories, typical carcinoid tumors, atypical carcinoid tumors (ACTs), Small cell carcinomas (SCNECs), and large cell neuroendocrine carcinomas (LCNECs), according to World Health Organization classifications. Well-differentiated typical and atypical carcinoids are categorized as low-grade neuroendocrine tumors (LGNETs), and the small and large cell types are categorized as high-grade neuroendocrine carcinomas (HGNECs). Small cell carcinoma is the most common type of NET, yet it accounts for less than 5% of all cervical carcinomas (Albores-Saavedra et al., 1997; Viswanathan et al., 2004). Because of the rarity of NETs, most studies on NETs have been perfomed on small samples or are case reports (Hoskins et al., 2003; Zivanovic et al., 2009). Therefore, the prognostic factors and treatment of patients with NETs are still controversial. We performed a retrospective review of 172 patients to analyze the clinicopathologic behaviors and prognostic factors of patients with NETs and to determine appropriate treatment. Lymph node metastasis was included in the revised 2018 International Federation of Gynecology and Obstetrics (FIGO) cervical cancer staging system. We evaluated the prognosis of NETs according to the FIGO 2018 staging system. We also evaluated the benefit of neoadjuvant therapy, adjuvant chemotherapy and adjuvant radiotherapy. Finally, we discuss the appropriate chemotherapy regimen and number of cycles for NETs.

Materials & methods

Patients and data collection

This was a retrospective study. After obtaining approval from the Ethics Committee of Fujian Cancer Hospital, Fujian, China (Ethical Application Ref: YKT2020-012-01), we reviewed the clinicopathological data of 195 patients with histologically confirmed NETs, who were diagnosed and treated at the Department of Gynecology of Fujian Cancer Hospital, Fujian, China between November 2002 and June 2019. All patients signed informed consent. Those who had incorrect pathology report or lacked follow-up data were excluded from the study. Histologic slides were reviewed by two pathologists specialized in gynecological cancers to confirm the diagnosis of neuroendocrine tumors of the uterine cervix by a central pathological review (CPR).

Central pathological review

The CPR was performed by two pathologists specializing in gynecological cancers. Biopsy specimens were obtained from patients who started treatment with neoadjuvant therapy and who did not undergo surgery, whereas hysterectomy specimens were obtained from patients who underwent surgery. The pathologists had a consensus on the criteria for the diagnosis of NETs according to World Health Organization classifications. Carcinoid tumors show prominent nucleoli with the nested, island, organoid, spindled, or trabecular pattern and are characterized by abundant cytoplasm, and characteristic granular chromatin. The differences between atypical carcinoid and carcinoid tumors are a greater degree of nuclear atypia and mitotic activity (5–10/10 HPF) as well as rare areas of necrosis. SCNEC is composed of ovoid, poorly cohesive cells, with condensed chromatin and scant cytoplasm. There is frequent nuclear molding, numerous mitotic figures, necrosis and apoptotic bodies. The growth pattern may be diffuse, trabecular, nested or exhibit rosette-like structures (Fig. 1A). LCNECs are recognized by their arrangement in well demarcated nests, trabeculae or cords with peripheral palisading. Tumor cells are large and polygonal, with vesicular or hyperchromatic nuclei and a prominent nucleolus. There is high mitotic activity and extensive geographic necrosis (Fig. 1B). Immunohistochemical staining for neuroendocrine markers, including CD56, chromogranin A or synaptophysin was performed to confirm neuroendocrine features (Figs. 1C & 1D). Nevertheless, positive neuroendocrine markers were not necessary for diagnosis (Conner et al., 2002). Mixed tumors were defined by the presence of squamous or glandular.

Figure 1 HE staining and immunohistochemistry of neuroendocrine tumors of the uterine cervix (A–D).

Hematoxylin and eosin staining showed the obvious morphological characteristics of small cell carcinoma and large cell carcinoma of the cervix. Immunohistochemistry showed that CD56, chromogranin A(CgA) and synaptophysin (Syn) were positive.

Treatment variables

The variables included age at diagnosis, tumor size, FIGO stage, pathological type, pure or mixed histology, lymph node metastasis, lymphovascular invasion (LVSI), parametrial involvement, perineural invasion, depth of stromal invasion, treatment and chemotherapy regimens and courses. lymph node metastasis was defined by histology in patients who underwent surgery, and by imaging studies, such as magnetic resonance imaging (MRI), computed tomography (CT), or positron emission computed tomography (PET), in patients who started treatment with neoadjuvant therapy or did not receive surgery.

We divided treatment into seven categories, surgery alone, surgery with adjuvant treatment, surgery preceded by neoadjuvant chemotherapy with or without adjuvant treatment, chemotherapy alone, radiotherapy alone, radiotherapy with chemotherapy and concurrent chemoradiotherapy.

Statistical analysis

The probabilities of overall survival (OS) and progression-free survival (PFS) were estimated using the Kaplan–Meier method. The Cox proportional hazards model was used to identify prognostic factors. Prognostic factors with P values < 0.1 in univariable analysis were further assessed in multivariable analysis. The hazard ratios (HR) and 95% confidence intervals (CI) were calculated using the Wald test. P values < 0.05 were considered statistically significant. The SPSS 24.0 statistical package (SPSS Inc., Chicago, IL, USA) was used for statistical analysis.

Results

Clinicopathologic characteristics of the patients

A total of 195 patients diagnosed with cervical neuroendocrine carcinoma were enrolled in our study between November 2002 and June 2019. The reasons for exclusion were as follows: three patients were diagnosed with other histology after the CPR, eight patients refused or discontinued treatment, five patients had insufficient medical information, and seven were lost to follow-up. Ultimately, 172 patients were enrolled in our study (Fig. 2).

Figure 2 Patients inclusion/exclusion process and treatment algorithm.

NACT, neoadjuvant chemotherapy; CCRT, concurrent chemoradiation; RT, radiotherapy; CT, chemotherapy; EP, etoposide and cisplatin/carboplatin; TP, paclitaxel and cisplatin/carboplatin; Other regimens, one patient received paclitaxel liposome, two received gemcitabine + platinum, one received bleomycin + ifosfamide + cisplatin, one received bleomycin + vincristine + cisplatin, one received temozolomide + xeloda, three received docetaxel + platinum, three received TP and EP successively.

The ages of the 172 enrolled patients ranged from 25 to 86 years, and the mean age was 46.7 years. The median cervical tumor dimension was 4.5 cm. There were 161 cases of small cell carcinoma, six cases of large cell carcinoma and four cases of typical carcinoid tumor tumors. One patient was classified as having a “NET, not classified” because both SCNEC and ACT components were present. Pure histology was documented in 73.3% (126/172) of the patients (Table 1). The mixed histology pattern was associated mostly with adenocarcinoma (29/172 patients; 16.7%). The remaining specimens were squamous cell carcinoma (14/172 patients; 8.1%) and adenosquamous carcinoma (3/172 patients; 1.7%). The median follow-up time was 50.7 months (range, 2–193).

Table 1 Patient, tumour, and treatment characteristics (N = 172).

Variables	No. of patients	%	
Hystological type	
Typical carcinoid tumor	4	2.3	
SCNEC	161	93.6	
LCNEC	6	3.5	
Not classified	1	0.6	
Histological homology	
Pure	126	73.3	
Mixed	46	26.7	
FIGO stage (2018)	
I	
IA	2	1.2	
IB1	4	2.3	
IB2	19	11.0	
IB3	6	3.5	
II	
IIA1	17	9.8	
IIA2	22	12.8	
IIB	27	15.7	
III	
IIIA	3	1.7	
IIIB	8	4.7	
IIIC1	38	22.1	
IIIC2	8	4.7	
IV	
IVA	0	0.0	
IVB	18	10.5	
FIGO stage (2018)	
IA-IIA2	71	41.3	
IIB-IVB	101	58.7	
Age (years)	
≤45	80	46.5	
>45	92	53.5	
Tumor size (cm)	
<2	9	5.2	
2–4	41	23.8	
4≤	122	70.9	
Lymph node metastasis	
Pelvic only	44	25.6	
Pelvic and para-aortic	15	8.7	
Negative	113	65.7	
Primary treatment	
Surgery + adjuvant therapy	36	20.9	
NACT + surgery ± adjuvant therapy	70	40.7	
Surgery alone	2	1.2	
CCRT + CT	38	22.1	
RT + CT	12	7.0	
CT alone	11	6.4	
RT alone	3	1.7	
Chemotherapy regimen	
EP	76	45.9	
TP	79	44.2	
Other regimens	12	7.0	
Without chemotherapy	5	2.9	
Note:

LGNET, low grade neuroendocrine tumor; HGNEC, high grade neuroendocrine carcinoma; SCNEC, small cell neuroendocrine carcinoma; LCNEC, large cell neuroendocrine carcinoma; NOT classified: one case included atypical carcinoid tumor and SCENC at the same time; Adjuvant therapy includes chemotherapy or radiotherapy; FIGO, International Federation of Gynecology and Obstetrics; NACT, neoadjuvant chemotherapy; CCRT, concurrent chemoradiation; CT, chemotherapy; RT, radiotherapy; Other regimens, one patient received paclitaxel liposome, two received gemcitabine + platinum, one received bleomycin + ifosfamide + cisplatin, one received bleomycin + vincristine + cisplatin, one received temozolomide + xeloda, three received docetaxel + platinum, three received TP and EP successively.

Of the 172 patients, 36 had stage I disease, 101 had stage II disease, 17 had stage III disease, and 18 had stage IV disease, based on the FIGO 2009 staging system. When the FIGO 2018 staging system was used, 31 patients were classified as stage I, 67 as stage II, 56 as stage III, and 18 as stage IV. Based on lymph node metastasis, 5 patients were classified as stage I, 34 patients as stage II, and 1 patient as stage IIIA were upstaged to stage IIIC according to the FIGO 2018 staging system. In this series, 108 patients received surgery as their primary treatment, and 53 patients received radiotherapy as their primary treatment. Among the 108 patients who received surgery, 105 received radical hysterectomy with pelvic lymph adenectomy and three received only simple hysterectomy without lymph node dissection. Overall (n = 172), 76 received etoposide and cisplatin/carboplatin (EP), 79 received paclitaxel and cisplatin/carboplatin (TP/TC) and 12 received the following regimens: one patient received paclitaxel liposomes, two received gemcitabine and platinum, one received bleomycin, ifosfamide and cisplatin, one received bleomycin, vincristine and cisplatin, onereceived temozolomide and Xeloda, three received docetaxel and platinum and three received TP and EP successively.

Survival outcomes

The median OS and PFS times were 33.23 months and 22.8 months, respectively, and the 5-year OS and PFS rates were 48.5% and 42.4%, respectively. At the end of follow-up (7th June 2020), 100 patients had experienced cancer recurrence and 92 patients had died. The 5-year survival rates of patients with stage I–IV disease were 74.8%, 56.2%, 41.4% and 0%, respectively. The 5-year progression-free survival rates of stage I–IV disease were 63.8%, 54.5%, 24.6%, and 0%, respectively. The survival curves for different FIGO stages are shown in Fig. 3. A comparison of OS and PFS revealed that the prognosis of stage I and II tumors classified based on the FIGO 2018 staging system tended to be better than that of stage I and II tumors classified based on the FIGO 2008 staging system, and the prognosis of stage III tumors classified with the 2018 staging system was worse than that of the corresponding tumors classified based on the FIGO 2008 staging system. However, these differences were not statistically significant.

Figure 3 Survival curves of all patients at each stage.

(A) OS and (B) PFS.

Among the entire series (n = 172), 100 patients experienced cancer recurrence, of which 98 had distant metastasis and 23 had local recurrence. The lung (50/98), liver (42/98), pelvic, and retroperitoneal, mediastinal and supraclavicular lymph nodes (32/98) were the most common sites of metastasis. In addition, four patients had brain metastases.

Prognostic factors

As data on the depth of parametrial extension, lymphovascular invasion, depth of stromal invasion, perineural invasion, neoadjuvant therapy and postoperative adjuvant radiotherapy were limited to patients who received surgery, these factors were not included in the univariate analyses of individuals with stages I–IV disease. We performed an additional analysis of the patients who underwent surgery. In the entire series (n = 172), multivariate analyses revealed that advanced FIGO stage (p = 0.006), age (≤45 vs. >45 years: 62.5% vs. 35.0%; p = 0.04) and tumor size (<4 vs. ≥4 cm: 76.0% vs. 39.0%; p = 0.013) were significant prognostic factors for OS. In addition, FIGO stage (p < 0.001), para-aortic lymph node metastasis (negative vs positive: 45.9% vs. 6.7%; p = 0.014) and larger tumor size (<4 vs. ≥4 cm: 64.9% vs. 33.7%; p = 0.015) were significant prognostic factors for PFS. The 5-year survival rates of patients with small cell carcinoma, large cell carcinoma and typical carcinoid tumors were 49.4%, 27.8%, 50.0%, and the 5-year disease-free survival rates were 43.2%, 33.3% and 25.0%, respectively. Histological type was not a prognostic factor. Moreover, histological homology, chemotherapy regimen and number of cycles of chemotherapy were not prognostic factors for NETs (Table 2). The survival curves for patients with different tumor sizes, ages, and para-aortic lymph node statuses are shown in Fig. 4.

Table 2 Prognostic factors for NETs of the uterine cervix (n = 172).

Variable	n	Overall survival	Progression free survival	
Univariate	Multivariate	Univariate	Multivariate	
5-year OS (%)	HR	95% CI	P	HR	95% CI	P	5-year PFS (%)	HR	95% CI	P	HR	95% CI	P	
Hystological typea			1.180	[0.708–1.967]	0.525					1.462	[0.864–2.473]	0.157				
Typical carcinoid tumor	4	50.0							25.0							
LCNEC	6	27.8							33.3							
SCNEC	161	49.4							43.2							
Age, years			1.754	[1.152–2.672]	0.009	1.569	[1.021–2.495]	0.040		1.521	[1.019–2.269]	0.04	1.441	[0.940–2.210]	0.094	
≤45	80	62.5							51.9							
>45	92	35.0							33.3							
Tumor size (cm)			3.198	[1.743–5.868]	<0.001	2.194	[1.179–4.082]	0.013		2.585	[1.533–4.360]	0.001	1.929	[1.133–3.282]	0.015	
<4	50	76.0							64.9							
≥4	122	39.0							33.7							
FIGO stage (2018)			2.053	[1.591–2.650]	<0.001	1. 726	[1.163–2.404]	0.006		2.067	[1.622–2.633]	0.001	1.731	[1.316–2.279]	<0.001	
I	31	74.8							63.8							
II	67	56.2							54.5							
III	56	41.4							30.8							
IV	18	0							0							
Histological homology			0.927	[0.569–1.509]	0.76					0.860	[0.543–1.361]	0.518				
Pure	126	47.4							40.2							
Mixed	46	52.5							49.1							
Pelvic LN metastasis			2.544	[1.681–3.85]	<0.001	1.381	[0.776–2.459]	0.273		2.799	[1.882–4.163]	<0.001	1.377	[0.772–2.456]	0.279	
No	113	59.9							55.2							
Yes	59	26.8							18.1							
Para-aortic LN metastasis			2.534	[1.406–4.567]	0.002	1.081	[0.549–2.127]	0.822		4.200	[2.411–7.317]	<0.001	2.185	[1.171–3.977]	0.014	
No	157	51.9							45.9							
Yes	15	16.0							6.7							
Chemotherapy regimen			1.221	[0.943–1.582]	0.129					1.145	[0.883–1.484]	0.307				
TP	79	52.1							41.9							
EP	76	49.2							45.6							
Other	12	30.5							23.8							
Without chemotheapy	5	40.0							40.0							
Cycle of chemotherapy			0.679	[0.439–1.051]	0.082	0.666	[0.423–1.049]	0.080		0.785	[0.521–1.181]	0.245				
0–5	107	45.2							40.9							
≥6	65	54.0							45.3							
Notes:

a One contains both SCNEC and ACT components.

LCNEC, large cell neuroendocrine carcinomas; SCNEC, Small cell neuroendocrine carcinoma; FIGO, International; Federation of Gynecology and Obstetrics; LN, lymphnode; EP, etoposide and cisplatin/carboplatin; TP, paclitaxel and cisplatin/carboplatin; Other regimens: one patient received paclitaxel liposome, two received gemcitabine + platinum, one received bleomycin + ifosfamide + cisplatin, one received bleomycin + vincristine + cisplatin, one received temozolomide + xeloda, three received docetaxel + platinum, three received TP and EP successively.

Figure 4 Comparison of survival curves in 172 patients with different tumor size, age and para-aortic lymph node status.

(A), (C) and (E) for OS, (B), (D) and (F) for PFS.

For the patients who underwent surgery (n = 108), pelvic lymph node metastasis was significantly associated with both OS and PFS in the multivariate analysis (Table 3). Those without pelvic lymph node metastasis had better 5-year OS (68.0% vs. 41.6%, p = 0.0028) and 5-year PFS (62.6% vs. 29.3%, p = 0.019) rates than those with pelvic lymph node metastasis. The survival curves for patients with different pelvic lymph node statuses are shown in Fig. 5. In addition, larger tumor size (<4 vs. ≥4 cm: 80.1% vs. 49.0%; p = 0.02) was a significant prognostic factor for OS. However, histological type, age, histological homology, parametrial involvement, lymphovascular invasion, depth of stromal invasion, perineural invasion, neoadjuvant therapy, adjuvant radiotherapy, chemotherapy regimen and number of cycles of chemotherapy were not prognostic factors (Table 3).

Table 3 Prognostic factors for patients who underwent surgical treatment (n = 108).

Variable	n	Overall survival	Progression free survival	
Univariate	Multivariate	Univariate	Multivariate	
5-year OS (%)	HR	95% CI	P	HR	95% CI	P	5-year PFS (%)	HR	95%CI	P	HR	95% CI	P	
Hystological typea			1.113	[0.505–2.453]	0.791					1.143	[0.514–2.544]	0.743				
Typical carcinoid tumor	2	50.0							50.0							
LCNEC	5	62.4							54.2							
SCNEC	100	33.3							40.0							
Age,years			1.612	[0.903–2.878]	0.106					1.564	[0.901–2.713]	0.112				
≤45	60	69.3							61.2							
>45	48	47.8							43.0							
Tumor size (cm)			2.901	[1.402–6.006]	0.004	2.440	[1.154–5.163]	0.020		2.176	[1.160–4.081]	0.015	1.938	[0.986–3.809]	0.055	
<4	43	80.1							68.9							
≥4	65	49.0							43.9							
FIGO stage (2018)			1.836	[1.222–2.759]	0.003	1.275	[0.587–2.770]	0.539		1.895	[1.271–2.825]	0.002	1.203	[0.576–2.512]	0.623	
I	29	72.3							64.5							
II	49	63.0							59.8							
III	29	43.8							32.7							
IV	1	0.0							0.0							
Histological homology			0.948	[0.513–1.753]	0.866					0.972	[0.549–1.722]	0.924				
Pure	68	59.9							51.3							
Mixed	48	61.8							57.4							
Pelvic LN metastasis			2.459	[1.368–4.420]	0.003	1.969	[1.076–3.603]	0.028		2.572	[1.471–4.499]	0.001	2.035	[1.126–3.676]	0.019	
No	77	68.0							62.6							
Yes	31	41.6							29.3							
Para-aortic LN metastasis			3.14	[0.426–23.137]	0.261					52.998	[4.806–584.481]	0.001	54.122	[4.458–657.123]	0.002	
No	107	61.3							53.8							
Yes	1	0.0							0.0							
Parametrial involvement			3.014	[1.075–8.452]	0.036	1.420	[0.477–4.231]	0.529		2.755	[0.99–7.666]	0.052	1.458	[0.495–4.294]	0.493	
Negative	103	62.6							54.9							
Positive	5	20.0							20.0							
Lymphovascular invasion			1.234	[0.696–2.186]	0.472					1.248	[0.724–2.151]	0.426				
Negative	61	63.6							58.6							
Positive	47	56.3							46.4							
Depth of stromal invasion			2.046	[1.058–3.956]	0.033	1.257	[0.587–2.770]	0.539		1.385	[1.009–1.901]	0.044	1.514	[0.773–2.967]	0.227	
Inner third	39	74.0							69.7							
Middle to outer third	69	52.9							44.2							
Perineural invasion			2.339	[0.559–9.783]	0.245					2.553	[0.615–10.600]	0.197				
Negative	105	61.7							54.0							
Positive	3	0.0							33.3							
Neoadjuvant therapy			0.826	[0.435–1.569]	0.559					0.817	[0.447–1.494]	0.511				
No	38	57.1							51.8							
Yes	70	61.2							53.5							
Adjuvant radiotherapy			1.885	[0.973–3.653]	0.060	1.559	[0.762–3.190]	0.224		1.672	[0.902–3.099]	0.103				
No	37	76.2							69.3							
Yes	71	52.4							44.3							
Adjuvant chemotherapy			1.184	[0.163–8.595]	0.867					1.471	[0.203–10.649]	0.702				
No	3	66.7							66.7							
Yes	105	60.4							53.0							
Chemotherapy regimen			1.009	[0.664–1.535]	0.965					1.037	[0.701–1.532]	0.857				
TP	49	58.0							53.3							
EP	49	66.1							56.0							
Other regimens	8	46.7							36.5							
Without chemotheapy	2	50.0							50.0							
Cycle of chemotherapy			0.764	[0.428–1.363]	0.362					0.823	[0.476–1.423]	0.485				
0–5	58	57.8							51.4							
≥6	50	63.7							55.8							
Notes:

a One contains both small cell carcinoma and atypical carcinoid tumor components.

LCNEC, large cell neuroendocrine carcinomas; SCNEC, Small cell neuroendocrine carcinoma; FIGO, International Federation of Gynecology and Obstetrics; LN, lymphnode; EP, etoposide and cisplatin/carboplatin; TP, paclitaxel and cisplatin/carboplatin; Other regimens: one received gemcitabine + platinum, one received bleomycin + ifosfamide + cisplatin, one received bleomycin + vincristine + cisplatin, three received docetaxel + platinum, one received TP and EP successively.

Figure 5 Comparison of survival curves in 108 patients who receiving surgery with different pelvic lymph node status.

(A) OS and (B) PFS.

Efficacy of treatment

In our study, patients in the early stage (stages I–IIA2) who received primary surgery tended to have better 5-year OS (67.8% vs. 44.4%, p = 0.199) and 5-year PFS (62.9% vs. 33.3%, p = 0.113) rates than those who received primary chemoradiation, but the difference was not statistically significant. The 5-year OS and 5-year PFS rates for patients with stage IIB–IIIC2 disease who received primary surgery were 50.6% and 34.7%, respectively, and the rates for patients who received primary radiotherapy were 42.4% and 41.3%, respectively. In patients with stage IV disease, primary treatment containing at least six cycles of chemotherapy was associated with significantly better 2-year OS (83.3% vs. 9.1%, p < 0.001) and 2-year PFS (57.1% vs. 0%, p = 0.01) rates than those primary treatment containing fewer than six cycles of chemotherapy (Table 4).

Table 4 Treatment and outcomes of the patients of NETs.

Treatment	N	5-year OS	P	5-year PFS	P	
Stage I–IIA2 (FIGO 2018)						
Primary surgery	62	67.8	0.199	62.9	0.113	
Primary RT	9	44.4		33.3		
Stage IIB–III (FIGO 2018)			0.673		0.285	
Primary surgery	45	50.6		34.7		
Primary RT	38	42.4		41.3		
Stages IVB (FIGO 2018)	N	2-year OS	P	2-year PFS	P	
Cycle of chemotherapy (0–5)	11	9.1	0.001	0.0	0.01	
Cycle of chemotherapy>5	7	83.3		57.1		
Note:

RT, radiotherapy.

Chemotherapy and toxicity

In our study, the chemotherapy regimen was not a significant prognostic factor for NETs. The 5-year OS rates of patients receiving the TP/TC regimen, EP regimen, other regimens and no chemotherapy were 52.1%, 49.2%, 30.5% and 40.0%, respectively. And the 5-year DFS rates of patients receiving the TP/TC regimen, EP regimen, other regimens and no chemotherapy were 41.9%, 45.6%, 23.8% and 40.0%, respectively. The differences were not statistically significant. We further studied the toxicity of the TP/TC regimen and EP regimen. The worst levels of toxicity reached at any time were recorded using the Common Terminology Criteria for Adverse Events (Version 5.0). The incidences of myelosuppression, hepatic dysfunction, and gastrointestinal reactions were 78.5% (62/79), 27.8% (22/79) and 13.9% (11/79), respectively, in patients who received the TP regimen and 75.0% (57/76), 22.4% (17/76), and 10.5% (8/76), respectively, in patients who received the EP regimen. Furthermore, the incidence of grade 3–4 toxicity was 45.6% (36/79) in patients who received the TP regimen and 44.7% (34/76) in patients who received the EP regimen. There were no significant differences in the above comparisons (Table 5). We believe that the two chemotherapy regimens have similar toxicity.

Table 5 The toxicities of TP and EP regimen.

Toxicity	TP	EP	χ2	P	
Myelosuppression			0.263	0.608	
No	17	19			
Yes	62	57			
Hepatic dysfunction			0.618	0.432	
No	57	59			
Yes	22	17			
Nausea/vomiting			0.416	0.519	
No	68	68			
Yes	11	8			
Grade 3–4 toxicities			0.011	0.917	
No	43	42			
Yes	36	34			

Discussion

To the best of our knowledge, this is the largest single-center retrospective study of neuroendocrine tumors of the uterine cervix. We studied the prognostic factors and outcomes of 172 patients with NETs based on the FIGO 2018 staging system. We discussed the appropriate primary treatment for each FIGO stage and evaluated the effect of adjuvant chemotherapy, adjuvant radiotherapy and neoadjuvant chemotherapy. In multivariate analysis, FIGO stage, older age and large tumors were independent prognostic factors for OS. In addition, FIGO stage, para-aortic lymph node metastasis and large tumors were independent prognostic factors for PFS. The TP/TC and EP regimens were the most commonly used regimens and had similar efficacy and toxicity.

FIGO stage and outcomes

At present, the prognostic factors for cervical neuroendocrine carcinoma are controversial; however, the FIGO stage is commonly recognized as an independent prognostic factor. In our study, the 5-year OS rates of patients with stage I–IV disease were 74.8%, 56.2%, 41.4% and 0%, respectively, according to the FIGO 2018 staging system, and the 5-year PFS rates were 63.8%, 54.5%, 30.8% and 0%, respectively. Wang et al. (2012) reported that the 5-year cancer-specific survival rate was 51.5% in patients with stage I–IIA disease (n = 123) and 24.9% in patients with stage IIB–IVB disease (n = 56). Intaraphet et al. (2014) reported a rate of 63% in patients with stage I disease (n = 71), 54% in patients with stage IIA disease (n = 11), 26% in patients with stage IIB disease (n = 26), and 0% in patients with stage III and IV disease (n = 22). The 5-year OS and 5-year PFS rates in our study were slightly better than those reported in previous studies. This better survival rate can be explained by our multimodality treatment. Upon the exclusion of 5 patients with stage IVB disease who received palliative chemotherapy, 98.2% (164/167) received two or more types of treatments. More than 97.1% (165/172) of patients with stage I–IV disease received chemotherapy, with an average of 4.35 cycles of chemotherapy. We believe that standard and complete multimodality treatment may improve the prognosis of NETs.

Prognostic factors

Older age was associated with poor survival in several studies (Chen, Macdonald & Gaffney, 2008; Intaraphet et al., 2014; Lee et al., 2008; Zhou et al., 2016b). Hoskins found that an age <50 years predicted prolonged OS (p = 0.02) (Hoskins et al., 2003). Our study revealed that patients aged 45 years or younger had a better survival rate than those aged older than 45 years (62.5% vs. 35.0%, p = 0.005). As shown in previous studies, large tumor size was an important prognostic factor for NETs (Atienza-Amores et al., 2014; Gardner, Reidy-Lagunes & Gehrig, 2011; Liao et al., 2012). Liao et al. (2012) conducted a large retrospective study of 293 patients and found that tumor size was indicative of a poor prognosis (≥4 cm vs. <4 cm, HR = 2.37, 95% CI [1.28–4.36], p = 0.006). Bermudez et al. (2001) reported that patients with tumors <4 cm had better 5-year OS rates than those with tumors >4 cm (76% vs. 18%, p < 0.05). Our study showed that large tumor size was a significant prognostic factor for both OS and PFS (p = 0.002 and p < 0.001). With more patients and studies, tumor size may ultimately prove to be an important prognostic factor for NETs.

NETs have a high incidence of lymph node metastasis, even early stage disease (Atienza-Amores et al., 2014; Satoh et al., 2014; Tempfer et al., 2018; Zhou et al., 2016a). Radical hysterectomy and pelvic lymph node dissection are commonly recommended in the primary treatment of patients with early stage NETs. However, few studies have discussed whether para-aortic lymph node dissection is essential. Boruta recommended radical surgery for all patients with early stage Nets, including para-aortic lymph node dissection (Boruta et al., 2001). In our study, 59 patients had pelvic lymph node involvement, and 15 patients had para-aortic node involvement at the initial diagnosis. After treatment and follow-up, 32 patients had pelvic, retroperitoneal, mediastinal or supraclavicular lymph node metastasis. Furthermore, we found that lymph node metastasis was an independent prognostic factor for both OS and PFS in patients with NETs who received surgery. Therefore, we recommend routine para-aortic lymph node dissection during radical surgery for patients with NETs, although more research is needed.

Primary treatment

Because of the rarity of NETs and the lack of multicenter randomized controlled studies, there is no standard treatment for NETs. Both the Gynecologic Cancer InterGroup (GCIG) and the Society of Gynecologic Oncology (SGO) recommend radical surgery as the primary treatment for patients with early stage disease, while chemoradiation is recommended for patients with advanced stage disease (Gardner, Reidy-Lagunes & Gehrig, 2011; Satoh et al., 2014). Most studies report long-term survival outcomes only for patients with early-stage disease who received radical surgical resection and adjuvant chemotherapy (Chen, Macdonald & Gaffney, 2008; Cohen et al., 2010; Lee et al., 2015; Zhou et al., 2016b). Cohen et al. (2010) showed that the 5-year disease-specific survival rate for patients with stage I–IIA disease (n = 169) who received radical hysterectomy was 38.2%, which was better than that for those who did not (23.8%) (p < 0.001). Ishikawa et al. (2018) found that the hazard ratio (HR) for death after definitive radiotherapy to death after radical surgery was 4.74 (95% confidence interval [CI], [1.01–15.90]). However, Chen et al. (2015) suggested that the survival outcomes after primary radiotherapy (RT) are superior to those after primary surgery. The authors attributed this hypothesis to the fact that surgical trauma may increase the number of circulating tumor cells (Khan et al., 2013). In our study, patients in the early stage (stages I–IIA2) who received primary surgery tended to have better 5-year OS rates and 5-year PFS rates than those who received primary chemoradiation, but the difference was not statistically significant. Primary surgery with adjuvant chemotherapy or chemoradiotherapy could be the optimal therapy for patients with early stage NETs.

Chemoradiotherapy is commonly recommended in the primary treatment of patients with advanced-stage NETs. Interestingly, we found that the survival outcomes of patients with advanced-stage disease who received primary radiotherapy were similar to those of patients who received primary surgery. The 5-year OS and 5-year PFS rates for patients with stage IIB–IIIC2 disease who received primary radiotherapy were 42.4% and 41.3%, respectively, and the rates for patients who received primary surgery were 50.6% and 34.7%, respectively. Surgery may also be an effective treatment for patients with advanced-stage NETs who are not sensitive to radiotherapy.

To the best of our knowledge, this is the first study to discuss the association between the number of cycles of chemotherapy and prognosis for patients with stage IV disease. In our study, primary treatment consisting of at least six cycles of chemotherapy was associated with significantly better 2-year OS (83.3% vs. 9.1%, p < 0.001) and 2-year PFS (57.1% vs. 0%, p = 0.01) rates than primary treatment consisting of fewer than six cycles of chemotherapy. Similar to small cell lung carcinoma, neuroendocrine cervical carcinoma is sensitive to chemotherapy, and chemotherapy can improve patient prognosis (Atienza-Amores et al., 2014; Gardner, Reidy-Lagunes & Gehrig, 2011; Satoh et al., 2014). Moreover, patients who received more than six cycles of chemotherapy in our study had larger tumor sizes (4.39 0.6 cm vs. 6.09 2.2 cm, p = 0.083). Larger tumors are usually associated with greater tumor burden and worse prognosis. In addition, the patients who received more than 6 cycles of chemotherapy were younger (52 12.3 years vs. 47.0 9.1, p = 0.329), but the difference was not statistically significant. Cohen found that chemotherapy (as primary treatment or adjuvant or with concurrent radiation) was associated with improved survival in stage IIB–IVA disease compared with treatment without chemotherapy (3-year survival: 17.8% vs. 12.0%; p = 0.043) (Cohen et al., 2010). Similarly, Wang reported that concurrent chemoradiation with EP for at least five cycles was associated with even better 5-year FFS (62.5% vs. 13.1%, p = 0.025) and CSS (75.0% vs. 16.9%, p = 0.016) than other treatments (Wang et al., 2012). An adequate number of cycles of chemotherapy seems to be indispensable for patients with stage IV disease. Due to the small number of cases in this study, further studies are needed to prove this finding.

Adjuvant treatment

Given the aggressive behavior of NETs, the use of adjuvant chemotherapy as a part of multimodality treatment has been commonly assessed. The 3-year distant recurrence-free survival rate was 83% for patients who received chemotherapy and 0% for patients who did not (p = 0.025) (Zivanovic et al., 2009). Intaraphet et al. (2014) reported a 5-year survival rate of 74.4% for patients who received adjuvant chemotherapy, a rate of 55.6% for those who received surgery alone, a rate of 53.3% for those who received adjuvant radiotherapy and a rate of 30.1% for those who received adjuvant chemoradiation (p = 0.041). In our study, adjuvant chemotherapy was used in most patients (n = 105) who underwent surgery (n = 108) and may be one of the reasons for their good prognosis.

In contrast, the benefit of adjuvant radiotherapy for NETs is controversial. Most authors have found that adjuvant radiotherapy does not improve survival. Lee et al. found that patients who received adjuvant radiotherapy had poorer 5-year survival rates than those who did not (45.5% vs. 52.5%, p = 0.37). Zhou et al. (2016b) reported that the 5-year CSS rates of patients who received radical surgery, radical surgery combined with radiotherapy, and radiotherapy alone were 67.9%, 49.7%, and 32.6%, respectively (p < 0.001). However, some authors have suggested that adjuvant radiotherapy reduces the local recurrence rate (Chen et al., 2015; Viswanathan et al., 2004). Viswanathan et al. (2004) reported that only 2 of 15 patients with NETs who received radiotherapy experienced recurrence in the radiation field, while 2 of 6 patients experienced recurrence after radical surgery without adjuvant radiotherapy. Owing to the aggressive behavior of these tumors, treatment failure usually results in distant metastasis rather than local recurrence (Atienza-Amores et al., 2014; Satoh et al., 2014). In our study, among the entire series (n = 172), 100 patients experienced cancer recurrence,of whom 98 had distant metastasis and 23 had local recurrence. Due to widespread hematogenous metastases, recurrence usually occurs outside the radiation field and may explain why adjuvant radiotherapy does not improve the outcomes of patients with NETs.

The GCIG and SGO recommend neoadjuvant chemotherapy (NAC) for patients with large tumors (>4 cm). However, few reports have indicated the benefit of neoadjuvant chemotherapy (Bermudez et al., 2001; Chang et al., 1999). Complete response has been observed in 6/7 patients with NETs after NAC (Chang et al., 1999), but the number of patients is so small that it is difficult to draw a definitive conclusion. Some authors found no improvement in overall survival among patients who received NAC (Dongol et al., 2014; Wang et al., 2012). In our study, neoadjuvant chemotherapy was not a prognostic factor. We found that the 70 patients who received NAC and the other 38 patients who did not receive NAC had similar 5-year OS and 5-year PFS rates. The 5-year OS rates were 61.1% and 57.1% (p = 0.559), respectively, and the 5-year PFS rates were 53.5% and 51.8% (p = 0.511), respectively. More studies are needed to prove the benefit of NAC.

Chemotherapy regimen

There is no standard chemotherapy regimen for NETs yet. Etoposide and cisplatin/carboplatin (EP) are the most commonly used regimens in the treatment of NETs (Atienza-Amores et al., 2014; Ishikawa et al., 2019; Lee et al., 2015; Pei et al., 2017; Tempfer et al., 2018). However, vincristine, Adriamycin and cyclophosphamide (VAC), irinotecan and platinum (CPT-P), paclitaxel and cisplatin/carboplatin (TC/TP) and various other chemotherapy regimens are also used. Pei et al. (2017) found that at least five cycles of adjuvant chemotherapy with EP (n = 39) was associated with better 5-year recurrence-free survival rates than other treatments (n = 46) (67.6% vs. 20.9%, p < 0.001). Zivanovic et al. (2009) reported that the 3-year recurrence-free survival rate was 83% for early stage patients who received chemotherapy and 0% for early stage patients who did not receive chemotherapy as part of their initial treatment (p = 0.025). However, Lei et al. reported that patients who received the TC regimen had better 5-year OS and 5-year DFS rates than those who received the non-TC regimen (p = 0.04). In our study, the chemotherapy regimen was not a significant prognostic factor for NETs. We found no significant difference in 5-year OS and 5-year PFS between patients receiving the TP/TC regimen and patients receiving the EP regimen or other regimens. More studies are urgently needed to find an appropriate chemotherapy regimen for NETs.

Conclusions

Because of the rarity of NETs, most studies on NETs have been perfomed with small samples or were case reports. To the best of our knowledge, three larger retrospective studies including 188, 193 and 179 patients have been published, exclusive of meta-analyses and studies based on the Surveillance, Epidemiology and End-Results (SEER) program (Cohen et al., 2010; Ishikawa et al., 2019; Wang et al., 2012). Our study is the largest single-center study with the fourth largest number of cases. We studied the prognostic factors and outcomes of NETs in 172 patients based on the FIGO 2018 staging system. We discussed the appropriate primary treatment for each FIGO stage and evaluated the efficacy of adjuvant chemotherapy, adjuvant radiotherapy and neoadjuvant chemotherapy. Moreover, we compared the chemotherapy regimens, and further analyzed the toxicities of the two most commonly used regimens. Notably, we found that patients with stage IV disease whose primary treatment included at least six cycles of chemotherapy had significantly better 2-year OS and 2-year PFS rates than those whose primary treatment included fewer than six cycles of chemotherapy. However, there are some limitations in this study. First, this study was retrospective; therefore, selection bias is unavoidable. Second, para-aortic LN metastasis was an important prognostic factor in our study, but only 12 of 108 patients underwent para-aortic lymph node dissection. Finally, this was a single-centre study, and the sample size was relatively small, especially regarding the number of LCNECs and carcinoids. More studies with larger cohorts are needed.

In conclusion, advanced FIGO stage, large tumor, older age and lymph node metastasis are independent prognostic factors for NETs of the cervix. Adjuvant chemotherapy, adjuvant radiotherapy and neoadjuvant chemotherapy did not improve the outcomes of patients with NETs. The TP/TC and EP regimens were the most commonly used regimens, with similar efficacies and toxicities. Standardized and complete multimodality treatment may improve the survival of patients with NETs.

Supplemental Information

Supplemental Information 1 Survival data in patients with different stage, tumor size, age and lymph node status.

Click here for additional data file.

Supplemental Information 2 Survival data in patients with stage IV.

Click here for additional data file.

We thank Xiaojie Wang for helpful suggestions and comments on the manuscipt.

Additional Information and Declarations

Competing Interests

Author Contributions

Human Ethics

Data Availability

The authors declare that they have no competing interests.

Jian Chen conceived and designed the experiments, performed the experiments, analyzed the data, prepared figures and/or tables, authored or reviewed drafts of the paper, and approved the final draft.

Yang Sun conceived and designed the experiments, prepared figures and/or tables, and approved the final draft.

Li Chen performed the experiments, prepared figures and/or tables, and approved the final draft.

Lele Zang performed the experiments, authored or reviewed drafts of the paper, and approved the final draft.

Cuibo Lin analyzed the data, authored or reviewed drafts of the paper, and approved the final draft.

Yongwei Lu analyzed the data, authored or reviewed drafts of the paper, and approved the final draft.

Liang Lin analyzed the data, authored or reviewed drafts of the paper, and approved the final draft.

An Lin analyzed the data, prepared figures and/or tables, and approved the final draft.

Hu Dan performed the experiments, prepared figures and/or tables, and approved the final draft.

Yiyu Chen performed the experiments, prepared figures and/or tables, and approved the final draft.

Haixin He conceived and designed the experiments, prepared figures and/or tables, and approved the final draft.

The following information was supplied relating to ethical approvals (i.e., approving body and any reference numbers):

The Ethics Committee of Fujian Cancer Hospital granted Ethical approval to carry out the study within its facilities (Ethical Application Ref: YKT2020-012-01).

The following information was supplied regarding data availability:

The survival data and the original data of the patients with stage IV disease are available in the Supplemental Files.

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
