# Peer review of "Prognostic factors and treatment of neuroendocrine tumors of the uterine cervix based on the FIGO 2018 staging system: a single-institution study of 172 patients"

_PeerJ, doi:10.7717/peerj.11563_

## Round 0.1 · original submission · Major Revisions

I would warmly recommend going through all the indications reported by the reviewers and paying attention to the correspondence between the text and the figures/tables.

Moreover, every statement reported in the Method section must be supported by evidence in the results and discussed in the appropriate section. Please do not report statements you cannot prove with data or argumentation.

Reviewer 1 ·

Basic reporting

I could not evaluate the article because there were discrepancies between the data described in the text and the Figures/Tables.
On line 182-190, authors described the multivariate analyses in the entire series using Figure 4 ant Table 2.
Figure 4 was comparison of survival curves and was not the result of multivariate analysis.
The title of Table 2 was prognostic factors (n¬¬=172). But in the Table 2, the total number of patients was not 172 and there was not the data of HR about FIGO stage, age and chemotherapy regimen that were described as the prognostic factors in the multivariate analyses.
The discrepancy was also seen on line 191 to 196.
Authors described about the multivariate analysis in the patients who underwent surgery using Figure 5 and Table 3.
But Figure 5 was not the data of multivariate analysis (just comparison of survival curves).
Table 3 was not the data of 108 patients because total number of patients was not 108.
Authors should carefully check the manuscript before submitting.

Experimental design

Authors described the efficacy of chemotherapy regimen using multivariate analyses. But the patients who did not receive chemotherapy should be included in the analysis.

Validity of the findings

Although the disease is rare, many observational studies have been conducted so far. No new discoveries are found in this paper.

Additional comments

Authors should carefully check the manuscript before submitting.

Reviewer 2 ·

Basic reporting

The number of cases is sometimes inconsistent throughout the manuscript. In line 137 in Results, the number of total patients is 195, but in line 88 in Methods, the number is 192.

Table 2 and Table 3 should be exchanged.

In Figure 1, the images are not small cell carcinoma but large cell neuroendocrine carcinoma. Please confirm the pathological diagnosis.

Experimental design

In Method, the authors classified neuroendocrine cervical tumors into carcinoid tumor, atypical carcinoid tumor (ACT), small cell carcinoma (SCNEC), and large cell neuroendocrine carcinoma (LCNEC). However, thereafter, they did not mention about that classification in text and in Tables. They should present clinical significance of the histological classification, i.e., ACT, SCNEC, and LCNEC, in Results and Tables 2 and 3.


In Table 2 and 3, the choice of parameters for multivariable analyses appears somewhat arbitrary. In Table 2, in overall survival analysis, they do not include parametrial involvement/depth of stromal invasion, and in progression-free survival analysis, they do not include tumor size, para-aortic lymph node metastasis, and depth of stromal invasion. In Table 3, in overall survival analysis, they do not include pelvic lymph node metastasis, para-aortic lymph node metastasis, and in progression-free survival analysis, they do not include age, pelvic lymph node metastasis.

Validity of the findings

In the section of 3.4 Efficacy of treatment, they mention that in patients with stage IV disease, primary treatment containing at least six cycles of chemotherapy was associated with significantly better 2-year OS (83.3% vs. 9.1%, P < 0.001) and 2-year PFS (57.1% vs. 0%, P = 0.01) rates than those primary treatment containing less than six cycles of chemotherapy. However, 5-year OS and 5-year PFS rates of all these groups were 0%. Such numbers appear unnatural. The reasons why the patients who received primary treatment containing <6 cycles of chemotherapy died within 2 years but a majority of patients who received primary treatment containing ≧6 cycles of chemotherapy did not suffer progression of cancers or death should be explained and discussed in detail in Discussion. For that purpose, for stage IV patients, they should also examine the data of therapeutic effect by RECIST and patients’ performance status.

In Discussion, line 228, the authors mention that the TP/TC and EP regimens were the most effective, with similar efficacies and toxicities. However, it is unclear with which regimen the authors compared the TP/TC and EP regimens. From Tables 2 and 3, chemotherapy regimens were classified into TP, EP, and others. However, the group of “others” is footnoted only as that including vincristine, bleomycin, Lipitor, idosfamide, temozolomide, and so on, and does not appear to be a single regimen. Some of them are not indication for cervical cancer or not anti-cancer agent. Therefore, it does not appear appropriate to compare TP and EP with the group of “others”.

Additional comments

Chen J. et al. studied prognostic factors and treatment of neuroendocrine tumors of the uterine cervix based on the FIGO 2018 staging system. They analyzed all stage patients together and drawn conclusion about prognostic factors and chemotherapeutic regimens to be recommended. However, the rationale of these recommendations is not sufficient from the present data. Especially, the conclusion that they recommend TP/TC and EP is not supported because the control against TP/TC and EP is the mixture of miscellaneous regimens to which detail is not described.

---

## Round 0.2 · Minor Revisions

I appreciate the work you have done to improve the clarity of your manuscript. Nevertheless, I believe you still need to revise it better for the language and the style.

In addition, pay attention to explain all the statements you make. Some of them lack clarity. For instance, in line 324 you state this is the first study to discuss the cycle of chemotherapy. What do you mean by this? Maybe the efficacy of the number of cycles?

Moreover, as one of the reviewers pointed out, there are other studies (not numerous, but available) on the topic, and I would like to have an idea of the size of your sample compared to the other. I am not asking for a review of the published studies, but it would be important for the reader to have an idea of where this study will be positioned in the current knowledge.

Finally, I would recommend adding a paragraph with the strength and limitations of your work.

Reviewer 2 ·

Basic reporting

.

Experimental design

.

Validity of the findings

.

Additional comments

Chen J et al. retrospectively examined prognostic factors and treatment in 172 patients with cervical neuroendocrine tumors (NETs) in a single institute. The findings that they described in this study were not of very high novelty and were not conclusive in a sense with regard to the effectiveness of therapies. Nonetheless, the analyses of a relatively large number of cases in a single institute appear to be worth reporting. The following comments are addressed for the improvement of contents as follows:

In Results 3.1, the authors compared survival between the 108 patients who received surgery as their primary treatment and 53 patients who received radiotherapy as their primary treatment. In Table 4, the number of patients who received primary surgery is 107 and that of patients who received primary radiotherapy was 47. Why did the authors exclude other 7 patients from the survival analyses?

Likewise, in Results 3.4 and Table 4, they compared OS and PFS between the patients who received treatment containing at least 6 cycles of chemotherapy and those who received treatment containing less than 6 cycles of chemotherapy in FIGO stage IV cases only. Why did not they compare the same parameters for Stage IIB-III patients and, if corresponding presents are present, for stage I-IIA2 patients?

The authors should discuss the reason why the prognosis of FIGO stage IV patients who received treatment containing less than 6 cycles of chemotherapy was extremely poor. The poor prognosis does not appear to have caused only by the insufficient amount of chemotherapy. There might be other major reasons that repelled the introduction or continuation of chemotherapy, for example, severe adverse effect, underlying systemic diseases, high performance status.

In Conclusion from line 390-391, they mention that adjuvant chemotherapy, adjuvant radiotherapy and neoadjuvant chemotherapy did not improve the outcomes of patients with NETs. However, they did not present data that support the statement/conclusion in this manuscript. They did not compare each treatment group with the control group.

In Figure 3, yellow curves are very hard to see. They should use other color or another pattern of black line.

Typographic errors, e.g., analysdis in line 132, dirrerent in legends for Figures 4 and 5.

In the legend of Figure 1, they did not present the image of chromogranin A (CgA). They should delete “chromogranin A (CgA)” from the legend.

---

## Round 0.3 · Minor Revisions

Thanks for the work you did to improve the text.

I would still recommend following the indication of the reviewer to adjust the consistency of the abstract.

Reviewer 2 ·

Basic reporting

The questions and comments have been answered appropriately.

Experimental design

The questions and comments have been answered appropriately.

Validity of the findings

Although there are some concern about the expression of the sentences in Abstract, they may be within a tolerable range. Examples of concern are: the 2nd line "to determine appropriate treatment guidelines"; and the last two lines "standardized and complete multimodality treatment may improve the survival of patients with NETs", which is somewhat inconsistent with the former sentence "adjuvant chemotherapy, adjuvant radiotherapy and neoadjuvant chemotherapy did not improve the outcomes of patients with NETs".

Additional comments

No comments.

---

## Round 0.4 · accepted · Accept

The authors have fulfilled all the requests.